# Scale-free non-Hermitian skin effect in a boundary-dissipated spin chain

He-Ran Wang, Bo Li, Fei Song and Zhong Wang

Institute for Advanced Study, Tsinghua University, Beijing, China

February 12, 2023

## Abstract

We study the open XXZ spin chain with a $PT$-symmetric non-Hermitian boundary field. We find an interaction-induced scale-free non-Hermitian skin effect by using the coordinate Bethe ansatz. The steady state and the ground state in the $PT$ broken phase are constructed, and the formulas of their eigen-energies in the thermodynamic limit are obtained. The differences between the many-body scale-free states and the boundary string states are explored, and the transition between the two at the isotropic point is investigated. We also discuss an experimental scheme to verify our results.

## 1 Introduction

Exactly solvable models play important roles in condensed matter physics, statistical physics, and mathematical physics. Certain experimentally relevant one-dimensional systems can be

modeled by open spin chains with boundary fields, some of which belong to the category of Yang-Baxter integrability. Examples include spin chains with diagonal [1–8] or off-diagonal [9–12] magnetic field. The problem is also related to classical dynamics of molecules with drain and source [13–17], and spin transport within the framework of Lindblad master equation [18–25]. Many mathematical tools, such as coordinate Bethe ansatz [1, 2, 6, 7, 14, 26], Sklyanin's reflection algebra (an open boundary version of algebraic Bethe ansatz) [3–5,8–12,27], matrix product operator ansatz [13, 15–19, 21–24], etc, have been developed to treat those systems.

In this article, we investigate a non-Hermitian open XXZ chain using coordinated Bethe ansatz. The chain is subjected to opposite imaginary magnetic field on two ends, pointing to a prescribed direction called $z$. Here, the non-Hermiticity naturally stems from the ubiquitous coupling with the environment. A special case has been studied thoroughly in previous literature, where the strength of the boundary field takes a specific value depending on the anisotropic interaction strength between adjacent spins. The Hamiltonian then respects the $q-$deformed $SU(2)$ symmetry [28] with $|q| = 1$, and serves as a representation of the Temperley-Lieb algebra [29,30]. The spectrum of the model is purely real, though the Hamiltonian is non-Hermitian. Furthermore, when $q$ is a root of unity (i.e., $q^n = 1$ for a certain integer $n$) for some values of the boundary field, the representation of the symmetry group enjoys richer structures, such that an exact duality exists between the spin model and free-end quantum Potts model. The duality leads to the same conformal field theory (CFT) structure of two models, thus the negative central charge obtained from the side of Potts model makes the non-Hermitian spin chain a typical example of recently introduced "non-unitary" CFT [31,32].

A significant consequence of $q$ being a root of unity is that the spin Hamiltonian develops Jordan blocks, which feature exceptional points. The number of Jordan blocks for given $q$ and size $N$ has been counted [33], followed by the constructions of the corresponding generalized eigenstates [34]. Given the existence of many-body exceptional points, it is natural to identify different phases around them. Our article exhausts the parameter space of boundary imaginary field and anisotropic interaction. For a small boundary field, the spectrum remains real, and the Bethe roots of the ground state only shift slightly compared with the Hermitian case. The spectrum becomes complex, however, when the boundary field exceeds the $q-$deformed $SU(2)$ symmetric value. We show that, despite the breaking of the quantum group invariance, the model possesses a novel behaviour of scale-free localization. We figure out the structure of the steady state (with the largest imaginary part of energy) and the ground state (with the lowest real part of energy) as the combination of a boundary Bethe root and a set of continuous Bethe roots in the thermodynamic limit. The continuous Bethe roots all have an imaginary part proportional inversely to the system size $N$, corresponding to a small imaginary wavevector (or momentum) $\kappa \sim \alpha/N$. In the single-particle context, when the localization length of wavefunction is proportional to the system size $N$, the density $|\psi_N(x)|^2 \sim \exp(2\alpha x/N)$ is invariant under re-scaling transformation with a factor $s$: $|\psi_N(x)|^2 = |\psi_{sN}(sx)|^2$, and therefore called scale-free non-Hermitian skin effect (NHSE) or critical NHSE [35–37]. The original NHSE means the exponential localization of most eigenstates near the boundary, with localization lengthes independent of the system size [38–44]; the scale-free NHSE is therefore a weaker version of NHSE. Scale-free NHSE has also been found in Hermitian systems with non-Hermitian boundary field, though the mechanism is different [45]. In the present work, the imaginary part of wavevector is attributed to the scattering between the boundary mode and magnons traveling in the bulk, and these Bethe roots contribute a non-negligible imaginary part to the energy. Thus, unlike previous works, our scale-free behaviour has a many-body origin. More precisely, it originates from the interplay between boundary dissipation and many-body interactions. On one hand, the interaction among magnons is an indispensable ingredient for the scale-free NHSE. On the other hand, we also note that the Hermitian counterpart, namely the open XXZ model subjected to real boundary field, has only isolated

boundary modes, and such continuous skin modes are lacking [6–8]. We derive an integral equation, dubbed imaginary Bethe equation, to solve the scale-free localization length in the thermodynamic limit. We then give an exact formula for the imaginary part of the steady state energy, which are then compared to finite-size numerical results. We also explain how to measure these physical quantities in cold-atom experiments.

Before proceeding, we compare our results to earlier studies on boundary-driven spin chains as open quantum systems. The evolution of those open quantum systems is generated by the Linbladian operator, composed of an integrable Hamiltonian and quantum jump operators on the boundary. A typical example relevant to our work is [19]

$$
\begin{aligned}
L(\rho) &= -i[H_{\text{XXZ}}, \rho] + \sum_{\mu=1,N} L_\mu \rho L_\mu^\dagger - \frac{1}{2}\{L_\mu^\dagger L_\mu, \rho\} \\
&= -iH_{\text{eff}}\rho + i\rho H_{\text{eff}}^\dagger + \sum_{\mu=1,N} L_\mu \rho L_\mu^\dagger,
\end{aligned}
$$

with $L_1 = \sqrt{g}S_1^-, L_N = \sqrt{g}S_N^+$ and $H_{\text{eff}} = H_{\text{XXZ}} - \frac{i}{2}\sum_{\mu=1,N} L_\mu^\dagger L_\mu$. Here, $H_{\text{eff}}$ is the non-Hermitian Hamiltonian we shall focus on below (see Eq. (1)). Although the Lindbladian breaks integrability, the density matrix of non-equilibrium steady state (NESS) has been established by the matrix product operator (MPO) ansatz exactly. Furthermore, it has been found that the local matrix of MPO ansatz is indeed the infinite-dimensional solution of Yang-Baxter relations, and thus exterior integrability emerges in the NESS [46, 47]. However, the dynamics towards NESS is unknown yet. Our work about the non-Hermitian effective Hamiltonian is complementary to the NESS solution because $H_{\text{eff}}$ governs the time evolution of the open quantum system under post-selection, which is relevant to numerous experiments [48]. Our solution is enabled by the Yang-Baxter integrability of the model. Another related system is the XXZ model with only one jump operator $L_1$ on the left boundary [26]. Since the dissipator is purely lossy, the Lindbladian becomes upper-triangular under an appropriate basis choice, so that the Liouvillian spectrum can be completely determined by the effective non-Hermitian Hamiltonian. The Hamiltonian has scale-free eigenstates even in the single-magnon sector, but $PT$ symmetry is absent due to that the dissipator occurs only on one of the two ends. By contrast, our Hamiltonian preserves $PT$ symmetry, and in single-magnon sector there are only Bloch-wave modes and exponentially localized states. Scale-free modes originate from many-body interactions in our model.

The rest part is organized as follows. In the next section, we introduce the model Hamiltonian, its general Bethe equations, and the phase diagram. In Sections 3.1 and 3.2, we consider the single-magnon and two-magnon state as a warm-up. We then generalize the results to the many-body cases to obtain the steady state with scale-free NHSE in Section 3.3. Section 3.4 is devoted to another type of steady state solution, the boundary string states, which emerges for the highly anisotropic case. In Section 4, we apply the ansatz of scale-free solutions to the ground state for different parameters. A possible experimental setup for the non-Hermitian model is discussed in Section 5. We give some concluding remarks in Section 6 .

## 2  Non-Hermitian XXZ model and the phase diagram

The Hamiltonian reads:

$$
H = -\sum_{j=1}^{N-1}(S_j^x S_{j+1}^x + S_j^y S_{j+1}^y + \Delta S_j^z S_{j+1}^z) + \frac{ig}{2}(S_N^z - S_1^z). \tag{1}
$$

where $S^\alpha = \frac{1}{2}\sigma^\alpha (\alpha = x, y, z)$ is the spin-1/2 operator; the anisotropic interaction strength $\Delta$ and boundary field strength $g$ are purely real, with $g > 0$. The unit circle $\Delta^2 + g^2 = 1$

has been the focus in previous literatures [1, 28–31] because it enjoys the $q$-deformed $SU(2)$ symmetry which greatly simplifies the problem. Here, we explore the entire $(\Delta, g)$ parameter space beyond this circle. We find new phenomena, including the interaction-induced scale-free non-Hermitian skin effect, that exist outside the unit circle.

The model respects the $PT$ symmetry with $TiT = -i$ and $PS_j^\alpha P = S_{N-j}^\alpha$, and therefore the eigenvalues are either real or form complex conjugate pairs. When the whole spectrum is purely real, the model is said to be in the $PT$ exact (or $PT$-symmetric) phase, otherwise it is in the $PT$ broken phase [49, 50]. The steady state, which has the largest imaginary part of eigen-energy in the $PT$ broken phase, is of great importance because it captures the long-time behaviour of the system. A generic initial wavefunction evolving for a sufficiently long time under $\exp(-iHt)$ will converge to the steady state; we shall study the phase diagram of this steady state. The Hamiltonian also commutes with total $z$ magnetization $m = \sum_{j=1}^N S_j^z$, so that it can be block diagonalized in each sector with definite total magnetization. Furthermore, there is another symmetry operator $PX$ with $X = \prod_{j=1}^N \sigma_j^x$ which sends $m$ to $-m$, and therefore it suffices to study non-positive magnetization ($m \leq 0$) states. For the odd length chain, $PX$ symmetry leads to the two-fold degeneracy of the steady states. Thus, we only take even site number $N$ throughout the paper.

Ferromagnetic ($\Delta > 0$) and anti-ferromagnetic ($\Delta < 0$) models can be related by the transformation $Z = \prod_j^{N/2} \sigma_{2j-1}^z$:

$$ZTH(\Delta, g)ZT = ZH(\Delta, -g)Z = -H(-\Delta, g). \tag{2}$$

As such, an eigenstate of ferromagnetic Hamiltonian with $H(\Delta, g)|\psi\rangle = E|\psi\rangle$ can be transformed to an eigenstate of the anti-ferromagnetic one: $H(-\Delta, g)(ZT|\psi\rangle) = -E^*(ZT|\psi\rangle)$. If $E$ has the largest imaginary part among the spectrum, so does $-E^*$. Thus, the steady state properties remain the same for $\pm\Delta$, and it suffices to study the ferromagnetic case.

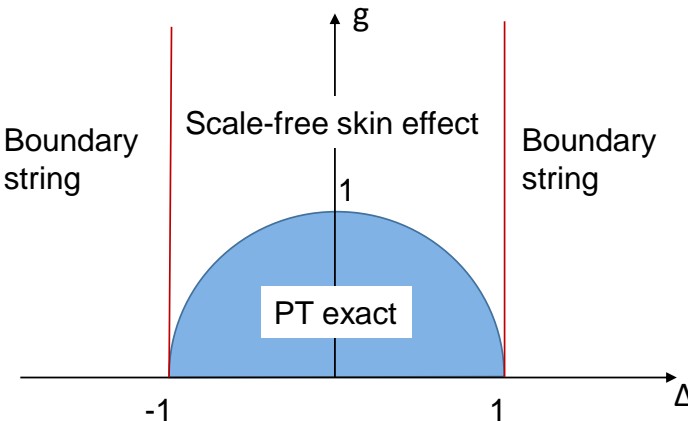

Figure 1: Steady-state phase diagram of model Eq. (1) in the zero magnetization sector. The critical curve $\Delta^2 + g^2 = 1$ separates $PT$ exact and broken phases. When $|\Delta| < 1$ and $g > g_c = \sqrt{1 - \Delta^2}$, the steady state is the many-body scale-free state; when $|\Delta| > 1$, the steady state is the boundary string state.

Eigenstates of the model can be solved by coordinate Bethe ansatz [1]. Take all spin down state as the reference state with energy $E_0 = -\frac{1}{4}(N-1)\Delta$, we can excite $M$ magnons by flipping $M$ spins up ($M \leq N/2$). We can construct the ansatz state $|\psi\rangle$, whose wavefunction in the

onsite magnon number basis is given by:

$$\langle n_1 \cdots n_j \cdots n_M | \psi \rangle = \sum_{\mathcal{P}} \sum_{\{\eta_j\}} (-1)^{\mathcal{P}} A(\{e^{ik_j}\}; \mathcal{P}, \{\eta_j\}) e^{i\eta_1 k_{\mathcal{P}1} n_1} \cdots e^{i\eta_j k_{\mathcal{P}j} n_j} \cdots e^{i\eta_M k_{\mathcal{P}M} n_M},$$

where $n_1 < \cdots < n_j < \cdots < n_M$ are the positions of up spin, $\mathcal{P}$ refers to all possible permutations, and chirality $\eta_j = \pm 1$ corresponds to right-moving and left-moving magnons. Relabeling the momentum of magnon by $\beta_j = e^{ik_j}$ [38, 51], we have the equivalent form:

$$\langle n_1 \cdots n_j \cdots n_M | \psi \rangle = \sum_{\mathcal{P}} \sum_{\{\eta_j\}} (-1)^{\mathcal{P}} A(\{\beta_j\}; \mathcal{P}, \{\eta_j\}) \beta_{\mathcal{P}1}^{\eta_1 n_1} \cdots \beta_{\mathcal{P}j}^{\eta_j n_j} \cdots \beta_{\mathcal{P}M}^{\eta_M n_M}. \tag{3}$$

The coefficient $A(\{\beta_j\}; \mathcal{P}, \{\eta_j\})$ is a function of magnon momenta $\{\beta_j\}$, permutation $\mathcal{P}$, and chirality $\{\eta_j\}$. Imposing the condition that $|\psi\rangle$ is an eigenstate of $H$ with energy

$$E_M(\{\beta_j\}) = E_0 + \sum_{j=1}^{M} (\Delta - (\beta_j + \beta_j^{-1})/2) \tag{4}$$

and the boundary condition, these coefficients can be found as [1]:

$$
\begin{aligned}
A(\{\beta_j\}; \mathcal{P}, \{\eta_j\}) \;=\; & \prod_{j}^{M} (1 - \Delta_+ \beta_{\mathcal{P}j}^{\eta_j}) \beta_{\mathcal{P}j}^{-(N+1)\eta_j} \\
& \prod_{0 \le k < l \le M} (1 - 2\Delta \beta_{\mathcal{P}l}^{\eta_l} + \beta_{\mathcal{P}l}^{\eta_l} \beta_{\mathcal{P}k}^{-\eta_k})(1 - 2\Delta \beta_{\mathcal{P}k}^{\eta_k} + \beta_{\mathcal{P}k}^{\eta_k} \beta_{\mathcal{P}l}^{\eta_l}) \beta_{\mathcal{P}l}^{-\eta_l},
\end{aligned}
$$

where $\Delta_{\pm} = \Delta \pm ig$. Here, the $M$ momenta have to satisfy the so-called Bethe equations:

$$\beta_j^{2(N-1)} \frac{(\beta_j - \Delta_+)(\beta_j - \Delta_-)}{(\beta_j^{-1} - \Delta_+)(\beta_j^{-1} - \Delta_-)} = \prod_{l \ne j}^{M-1} \frac{1 - 2\Delta\beta_j + \beta_j\beta_l}{1 - 2\Delta\beta_l + \beta_j\beta_l} \frac{1 - 2\Delta\beta_j + \beta_j\beta_l^{-1}}{1 - 2\Delta\beta_l^{-1} + \beta_j\beta_l^{-1}}. \tag{5}$$

Intuitively, the left hand side is the phase accumulated by a magnon when it travels freely from one end of the chain to the other and then gets reflected back; each term of the right hand side represents the scattering phase between magnon $j$ and $l$. Note that for the periodic XXZ model there is only the $\beta_j, \beta_l$ term in the scattering phase, while for the open boundary condition $\beta_j$ also scatter with $\beta_l^{-1}$. The solutions are inversion-symmetric in the sense that $\beta_j$ and $\beta_j^{-1}$ always appear in pairs in the solutions. After solving a set of Bethe roots $\{\beta_j\}$, the energy of $M$ magnon state is Eq. (4). Notably, if $|\beta_j| = 1$ ($j = 1, \cdots, M$), i.e., all $\beta_j$'s belong to the unit circle, one must have $\text{Im}(E_M(\{\beta_j\})) = 0$. Thus, PT symmetry breaking requires $|\beta_j| \ne 1$ for certain $\beta_j$'s, which implies the presence of NHSE (which turns out to be scale-free here). Although it remains unclear whether all eigenstates of the non-Hermitian XXZ model have the form of Bethe ansatz, it turns out that the ground state and the steady state can be solved exactly.

    Our results on the steady states for different parameters are summarized in Fig. 1. We focus on the zero magnetization sector ($m = 0$) since for the Hermitian $XXZ$ model (with $\Delta < 1$) the ground state lies in this sector. Our numerical results support that the steady state belongs to the zero magnetization sector. The phase boundary between $PT$-exact and broken phase is $\Delta^2 + g^2 = 1$. Given the transformation between $H(\Delta, g)$ and $H(-\Delta, g)$, the steady-state phase diagram is symmetric about the $g$ axis. Apart from the many-body scale-free modes which will be investigated in detail in Section 3, we identify the "boundary string" state as the steady state when $|\Delta| > 1$. A boundary string corresponds to a multi-magnon bound state localized at the boundary, which will be explained in Section 3.4.

## 3 Bethe ansatz solutions for scale-free skin modes

### 3.1 Single-magnon state

In the single-magnon sector ($M = 1$), the Bethe equation (5) is simplified to

$$\beta^{2(N-1)} \frac{(\beta - \Delta_+)(\beta - \Delta_-)}{(\beta^{-1} - \Delta_+)(\beta^{-1} - \Delta_-)} = 1. \tag{6}$$

When $|\Delta_\pm| < 1$, or equivalently $g < g_c$, solutions of the equation have been found to be on the unit circle [52]. We briefly review the proof. We define complex variable function $h(\beta) : \mathbb{C} \to \mathbb{C}$ as

$$h(\beta) = \beta^{N-1}(\beta - \Delta_+)/(\beta^{-1} - \Delta_-).$$

Eq. (6) is then transformed to $h(\beta) = h(\beta^{-1})$. For $g < g_c$, the image of the disk $|\beta| < 1$ under $h$ is still inside the disk, that is, $|h(\beta)| < 1$, and vice versa. The statement can be verified by writing $\beta = \rho \exp(i\phi), \Delta_+ = \rho_0 \exp(i\phi_0)$ with $\rho_0 < 1$, then

$$|\beta - \Delta_+|^2 - |\beta^{-1} - \Delta_-|^2 = (\rho - \rho^{-1})(\rho + \rho^{-1} - 2\rho_0 \cos(\phi - \phi_0)).$$

Since $\rho + \rho^{-1} \geq 1 > 2\rho_0 \cos(\phi - \phi_0)$, we have $|\beta - \Delta_+| < |\beta^{-1} - \Delta_-|$ when $|\beta| < 1$ ($\rho < 1 < \rho^{-1}$), therefore $|h(\beta)| < 1$. A similar argument works for $|\beta| > 1$. Thus, possible solutions of $h(\beta) = h(\beta^{-1})$ must be on the unit circle, corresponding to purely real momentum.

When $g > g_c$, no theorem prohibits the existence of non-unitary solutions, and one can notice that a pair of isolated boundary modes with $\beta_\pm \approx \Delta_\pm$ is possible. For such a solution, $|\beta_\pm| > 1$ leads to the divergence of the term $\beta^{2(N-1)}$ in Eq. (6) in the thermodynamic limit, but this can be compensated by the factor $(\beta - \Delta_+)(\beta - \Delta_-)$, which is close to zero. Most of the energy levels remain real, and the scale-free localization behavior is absent in this sector. We define the boundary imaginary energy contributed by one boundary mode as

$$E_b = -\frac{1}{2} \text{Im}(\Delta_- + \Delta_-^{-1}) = \frac{1}{2}(g - \frac{g}{\Delta^2 + g^2}). \tag{7}$$

### 3.2 Two-magnon state

In the $M = 2$ sector, scale-free modes appear and contribute to the $1/N$ scaling behaviour of energy. Fig. 2(a1) shows a typical finite-size two-magnon spectrum. We color the eigenvalues by many-body participation entropy [53–55], a measure of localization generalized from single-particle inverse participation ratio (IPR):

$$S_2(|\psi\rangle) = -\log \frac{\sum_i |\psi_i|^4}{(\sum_i |\psi_i|^2)^2},$$

where the index $i$ sums over all the basis functions in the relevant Hilbert space, $|\psi\rangle$ is a many-body eigenstate. We choose $|i\rangle$ to be local magnon number basis for the following calculations. The participation entropy gets smaller when the eigenstate is more localized in the Hilbert space, as we observed in Fig. 2(a1): Two dark points correspond to isolated states bounded to the boundaries, and both two magnons are localized to the same side; around $\text{Im}(E) = \pm E_b$ there is a continuum of states, which are the combination of a boundary mode with $\beta_1 \approx \Delta_\pm$ and a scale-free mode with $\beta_2 \approx e^{ik}$; the continuum on the real axis is brighter, though there is one localized mode, corresponding to the state with two magnons localized on different ends.

We apply Bethe equation to the state with one magnon bounded to the boundary while the other has momentum $\beta_2$:

$$|\beta_2|^{2N} \approx |\frac{1 - 2\Delta\beta_2 + \Delta_-\beta_2}{1 - 2\Delta\Delta_- + \Delta_-\beta_2} \frac{1 - 2\Delta\beta_2 + \Delta_-^{-1}\beta_2}{1 - 2\Delta\Delta_-^{-1} + \Delta_-\beta_2}|. \tag{8}$$

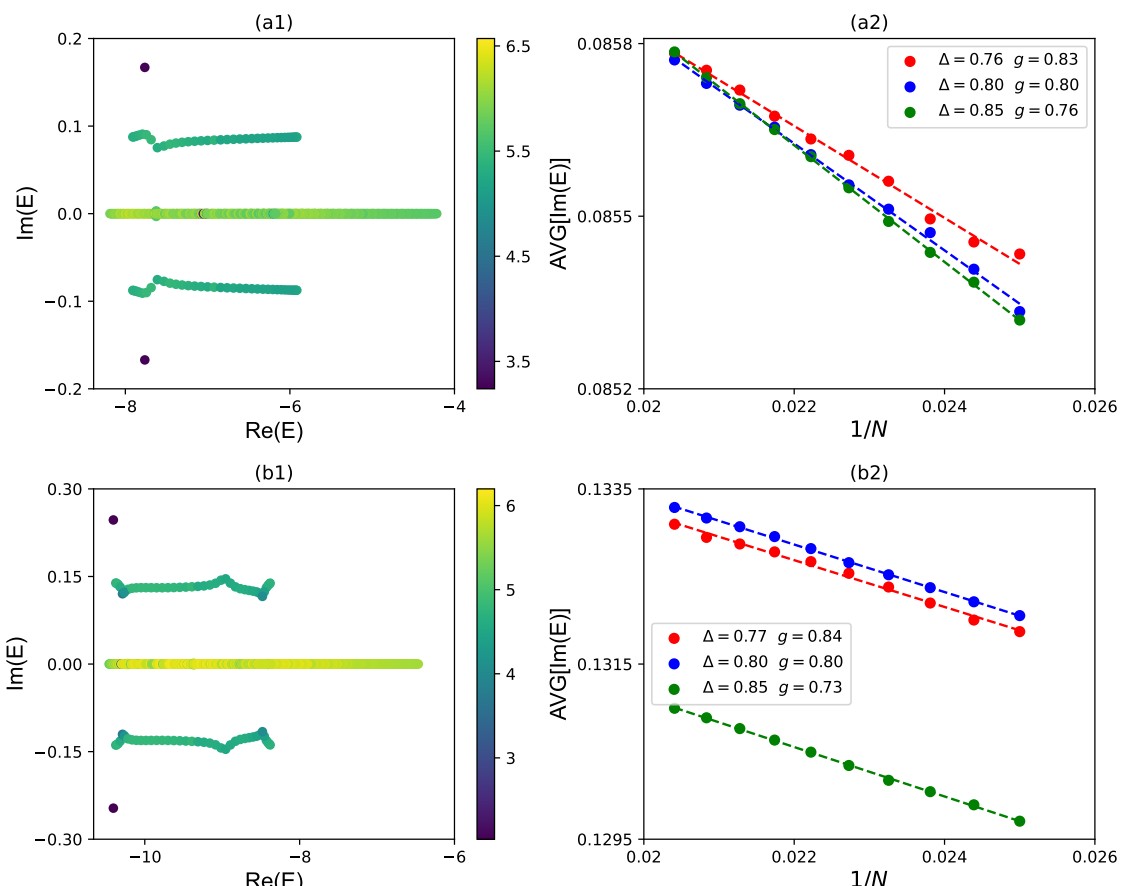

Figure 2: (a1) and (b1) The spectrum of $M = 2$ sector for $N = 40, \Delta = 0.8, g = 0.8$. Eigenvalues are colored by the participation entropy of the corresponding eigenstate. (a2) and (b2) Finite-size scaling for the average of the imaginary part of the energy. Only those states with $\text{Im}(E) > 0$ in the continuous spectrum are taken into consideration. (a1) and (a2) are based on the pristine model eq. (1); for (b1) and (b2), additional next-nearest-neighbor coupling $\Delta' = 0.3$ is added [see eq. (10)].

The right hand side, which will be denoted by $\exp(2\kappa)$, is of order $\mathcal{O}(1)$. Taking logarithm of both sides, we have $2N\ln|\beta_2| = 2\kappa + \mathcal{O}(1/N)$. Thus, $\ln\beta_2$ acquires a first order correction to the real part:

$$\text{Re}(\ln\beta_2) = \ln|\beta_2| = \kappa/N + \mathcal{O}(1/N^2).$$

Traveling along the chain, the corresponding magnon accumulates an amplitude change $A = |\beta_2|^N \sim \exp(\kappa)$. This magnon localization is distinguished from the disorder-induced Anderson localization and the original NHSE, both of which have exponential eigenstate decay $\psi(x) \sim \exp(\kappa' x)$ with size-independent localization lengthes, so that the wavefunction amplitude change ($\sim \exp(\kappa' N)$) diverges as the system size $N$ grows to infinity. In contrast, the amplitude change $A$ in our case saturates as the size $N$ grows. In terms of the generalized Brillouin zone (GBZ) of the non-Bloch band theory [38, 51], $\ln|\beta_2| \approx \kappa/N$ means that the GBZ (more precisely, the finite-size GBZ [56]) deviates from the unit circle by amount $\mathcal{O}(1/N)$. Notably, the scale-free localization in our model has an intrinsic many-body origin because in the non-interacting limit $\Delta = 0$, the right hand side of Eq. (8) equals 1, and the imaginary momentum vanishes. Therefore, the phenomenon in our model is dramatically different from that of free-particle models [35–37].

The energy of such two-magnon state is

$$\text{Im}(E) = E_b - \frac{1}{2}\text{Im}(\beta_2 + \beta_2^{-1}) = E_b - \frac{1}{N}\kappa\sin(k). \tag{9}$$

The statement is verified by finite-size scaling of the average of those complex energy. As shown in Fig. 2(b1), the imaginary part scales linearly with $1/N$. In the thermodynamic limit, it will converge to $\pm E_b$. Moreover, we add a next-nearest-neighbor $zz$ interaction

$$H_{nn} = -\sum_{j=1}^{N-2}\Delta' S_j^z S_{j+2}^z \tag{10}$$

to break integrability, yet the numerical results [Fig. 2(b1)(b2)] show no qualitative differences, which supports the universality of scale-free behaviour.

## 3.3 Imaginary Bethe equation at $0 < \Delta < 1$

After the warm-up on two-body scale-free modes, we now generalize it to the many-body cases. We will study the parameter space $0 < \Delta < 1$, where it is the Luttinger liquid phase in the Hermitian limit and the ground state lies in $M = N/2$ sector (with zero magnetization). We assume that the steady state is composed of a boundary mode and a set of continuous scale-free Bethe roots, and then derive the Bethe equations in the thermodynamic limit.

We adopt a conventional parametrization of magnon momentum [57, 58]:

$$\beta_j = -\frac{\sinh[\gamma(x_j - i)/2]}{\sinh[\gamma(x_j + i)/2]}, \tag{11}$$

where $\gamma = \arccos(-\Delta)$ such that $0 < \gamma < \pi$. The kinetic energy of the magnon is

$$E(x_j) = -\frac{1}{2}(\beta_j + \beta_j^{-1}) = \frac{1 - \cos(\gamma)\cosh(\gamma x_j)}{\cos(\gamma) - \cosh(\gamma x_j)}. \tag{12}$$

Taking the logarithm of Bethe equations (5), we have

$$2N\theta_1(x_j) + \phi_b(x_j) = 2\pi I_j + \sum_{l\neq j}[\theta_2(x_j - x_l) + \theta_2(x_j + x_l)], \tag{13}$$

where the function $\theta_n$ is defined as

$$\theta_n(x) = 2\arctan[\cot(n\gamma/2)\tanh(\gamma x/2)]. \tag{14}$$

The second term $\phi_b(x_j)$ on the left of Eq. (13) is the scattering phase between the magnon $j$ and the boundary, which has the form:

$$\phi_b(x) = \theta_{m_+}(x) + \theta_{m_-}(x),$$

where $\theta_{m_\pm}$ is obtained by taking $n$ in Eq. (14) as complex numbers

$$m_\pm = \frac{1}{\gamma}\theta_1\left(\frac{\ln(\cos(\gamma) \pm ig)}{\gamma}\right) + \frac{i\pi}{2\gamma}.$$

This involved boundary term will not have significance in the rest part of solution. For the right hand side, $I_j$ is an integer and the set of $\{I_j\}$ determines the set of Bethe roots. We take

the occupation of one boundary mode and $I_j = j$ on the steady state, and the corresponding boundary Bethe root $\beta_0 = \Delta_-$ has the parametrization

$$x_0 = \frac{1}{\gamma} \ln \frac{g - \sin(\gamma)}{g + \sin(\gamma)} - i = \lambda_0 - i. \tag{15}$$

The steady-state Bethe equations becomes

$$2N\theta_1(x_j) + \phi_b(x_j) = 2\pi j + \sum_{l \neq 0,j}[\theta_2(x_j - x_l) + \theta_2(x_j + x_l)] + \theta_2(x_j - x_0) + \theta_2(x_j + x_0). \tag{16}$$

We rewrite $x_j = \lambda_j + i\sigma_j/N$ with purely real $\lambda_j, \sigma_j$. We note that $|\sigma_j/N| \ll \lambda_j$, and therefore any real function of $x_j$ can be expanded as

$$f(x_j) = f(\lambda_j) + if'(\lambda_j)\sigma_j/N + \mathcal{O}(1/N^2). \tag{17}$$

The real and imaginary part of Bethe equations are:

$$2N\theta_1(\lambda_j) + \phi_b(\lambda_j) + \mathcal{O}(1) = 2\pi j + [\sum_{l \neq 0,j} \theta_2(\lambda_j - \lambda_l) + \theta_2(\lambda_j + \lambda_l)]$$
$$+ \mathrm{Re}[\theta_2(\lambda_j - x_0) + \theta_2(\lambda_j + x_0)], \tag{18}$$

$$(2\theta_1'(\lambda_j) + \frac{1}{N}\phi_b'(\lambda_j))\sigma_j + \mathcal{O}(\frac{1}{N}) = \frac{1}{N}\sum_{l \neq 0,j} \theta_2'(\lambda_j - \lambda_l)(\sigma_j - \sigma_l) + \theta_2(\lambda_j + \lambda_l)(\sigma_j + \sigma_l)]$$
$$+ \mathrm{Im}[\theta_2(\lambda_j - x_0) + \theta_2(\lambda_j + x_0)]. \tag{19}$$

Eq. (18) and Eq. (19) is accurate only to the $\mathcal{O}(N)$ and $\mathcal{O}(1)$ order, respectively, and their leading terms are:

$$2N\theta_1(\lambda_j) = 2\pi j + [\sum_{l \neq 0,j} \theta_2(\lambda_j - \lambda_l) + \theta_2(\lambda_j + \lambda_l)], \tag{20}$$

$$2\theta_1'(\lambda_j)\sigma_j = \frac{1}{N}\sum_{l \neq 0,j} \theta_2'(\lambda_j - \lambda_l)(\sigma_j - \sigma_l) + \theta_2(\lambda_j + \lambda_l)(\sigma_j + \sigma_l)]$$
$$+ \mathrm{Im}[\theta_2(\lambda_j - x_0) + \theta_2(\lambda_j + x_0)]. \tag{21}$$

Eq. (21) indicates that each scattering between the $(j, l)$ magnon pair generates a $\mathcal{O}(1/N)$ contribution to $\sigma_j$, and therefore the sum over $l$ is of order $\mathcal{O}(1)$. Since a nonzero $\sigma_j \sim \mathcal{O}(1)$ characterizes the scale-free NHSE (with imaginary part of momentum $\sim \sigma_j/N$), Eq. (21) clearly demonstrates the many-body origin of the scale-free NHSE in the present model.

Eq. (20) is the same as the ground state Bethe equations of the Hermitian open XXZ model [1]. It is standard to calculate the difference between $(j+1)$−th and the $j$−th equation, taking $f(\lambda_{j+1}) - f(\lambda_j) = f'(\lambda_j)(\lambda_{j+1} - \lambda_j)$:

$$2N\theta_1'(\lambda_j)(\lambda_{j+1} - \lambda_j) = 2\pi + [\sum_{l \neq 0,j} \theta_2'(\lambda_j - \lambda_l) + \theta_2'(\lambda_j + \lambda_l)](\lambda_{j+1} - \lambda_j). \tag{22}$$

The thermodynamic limit is taken by sending

$$\lim_{N \to \infty} \frac{1}{N(\lambda_j - \lambda_{j+1})} = \rho(\lambda_j), \qquad \lim_{N \to \infty} \frac{1}{N}\sum_l f(\lambda_l) = \int_0^\infty d\lambda \rho(\lambda)f(\lambda), \tag{23}$$

then the integral equation of $\rho(\lambda)$ is

$$\frac{1}{2}\rho(\lambda) + \int_{-\infty}^{+\infty} d\lambda' \frac{K_2(\lambda - \lambda')}{2\pi} \frac{1}{2}\rho(\lambda') = \frac{K_1(\lambda)}{2\pi}, \tag{24}$$

where $K_n(\lambda)$ is the derivative of $\theta_n(\lambda)$ with respect to $\lambda$:

$$K_n(\lambda) = \theta'_n(\lambda) = \frac{\gamma \sin(n\gamma)}{\cosh(\gamma\lambda) - \cos(n\gamma)}.$$

Notably, the energy function is proportional to $K_1$ up to a constant:

$$E(\lambda) = -\sin(\gamma)K_1(\lambda)/\gamma + \cos(\gamma). \tag{25}$$

Note that only when the steady state belongs to the zero magnetization sector, the integral interval of $\lambda$ can be taken as $(-\infty, \infty)$. This is the case here, and it corresponds to filling the Fermi sea $k \in (-\pi + \gamma, \pi - \gamma)$. The integral equation (24) is commonly solved by Fourier transformation:

$$
\begin{aligned}
\tilde{\rho}(\omega) &= \int_{-\infty}^{+\infty} \frac{d\lambda}{2\pi} \rho(\lambda) e^{i\omega\lambda}, \\
\tilde{K}_n(\omega) &= \int_{-\infty}^{+\infty} \frac{d\lambda}{2\pi} K_n(\lambda) e^{i\omega\lambda} = \frac{\sinh(\frac{\pi}{\gamma} - n)\omega}{\sinh \frac{\pi}{\gamma}\omega}.
\end{aligned}
\tag{26}
$$

Applying the convolution formula on Eq. (24), we have a linear equation

$$\frac{1}{2}(1 + \tilde{K}_2(\omega))\tilde{\rho}(\omega) = \frac{\tilde{K}_1(\omega)}{2\pi}, \tag{27}$$

then the distribution function is solved: $\tilde{\rho}(\omega) = 1/(2\pi\cosh\omega), \rho(\lambda) = 1/(2\cosh(\pi\lambda/2))$.

Eq. (21) counts two kinds of mechanisms of scale-free localization. On the right hand side, the first term sums interactions between magnons in the bulk, while the second term is the scattering with the boundary mode. The last one is much smaller than the first in a few-body state, but becomes comparable when the magnon number is the same order of system size $N$, e.g. in the zero magnetization sector. Define $\sigma(\lambda)$ as a function of $\lambda$, the continuous version of this equation is

$$\rho(\lambda)\sigma(\lambda) + \int_{-\infty}^{+\infty} d\lambda' \frac{K_2(\lambda - \lambda')}{2\pi} \rho(\lambda')\sigma(\lambda') = \frac{1}{2\pi}\text{Im}[\theta_2(\lambda - x_0) + \theta_2(\lambda + x_0)]. \tag{28}$$

Dubbed "imaginary Bethe equation", it is a central result of this article. To derive the linear equation of $\tilde{\sigma}_\rho(\omega) = \int_{-\infty}^{+\infty} \frac{d\lambda}{2\pi} e^{i\omega\lambda} \rho(\lambda)\sigma(\lambda)$, we need to take the Fourier transformation. The left hand side reads $(1 + \tilde{K}_2(\omega))\tilde{\sigma}_\rho(\omega)$, while the right hand side is:

$$
\begin{aligned}
&\int_{-\infty}^{+\infty} \frac{d\lambda}{2\pi} e^{i\omega\lambda} \text{Im}[\theta_2(\lambda - x_0) + \theta_2(\lambda + x_0)] \\
=&\int_{-\infty}^{+\infty} \frac{d\lambda}{2\pi} e^{i\omega\lambda} \text{Im}[\theta_2(\lambda - \lambda_0 + i) + \theta_2(\lambda + \lambda_0 - i)] \\
=&2i\sin(\omega\lambda_0) \int_{-\infty}^{+\infty} \frac{d\lambda}{2\pi} e^{i\omega\lambda} \text{Im}[\theta_2(\lambda + i)] \\
=&-\frac{2\sin(\omega\lambda_0)}{\omega} \int_{-\infty}^{+\infty} \frac{d\lambda}{2\pi} e^{i\omega\lambda} \frac{d}{d\lambda}\text{Im}[\theta_2(\lambda + i)] \\
=&\frac{2\sin(\omega\lambda_0)}{\omega} \int_{-\infty}^{+\infty} \frac{d\lambda}{2\pi} e^{i\omega\lambda} \frac{2\gamma\cos(\gamma)\sin^2(\gamma)\sinh(\gamma\lambda)}{(\cosh(\gamma\lambda) - \cos(\gamma))(\cosh(\gamma\lambda) - \cos(3\gamma))} \\
=&2i\sin(\omega\lambda_0)\sinh(\omega) \frac{\sinh[(\frac{\pi}{\gamma} - 2)\omega]}{\omega\sinh(\frac{\pi}{\gamma}\omega)}.
\end{aligned}
\tag{29}
$$

Denoting the above expression by $\Theta(\omega)$, we can solve $\tilde{\sigma}_\rho(\omega)$:

$$\tilde{\sigma}_\rho(\omega) = \frac{1}{1+\tilde{K}_2(\omega)} \frac{\Theta(\omega)}{2\pi}. \tag{30}$$

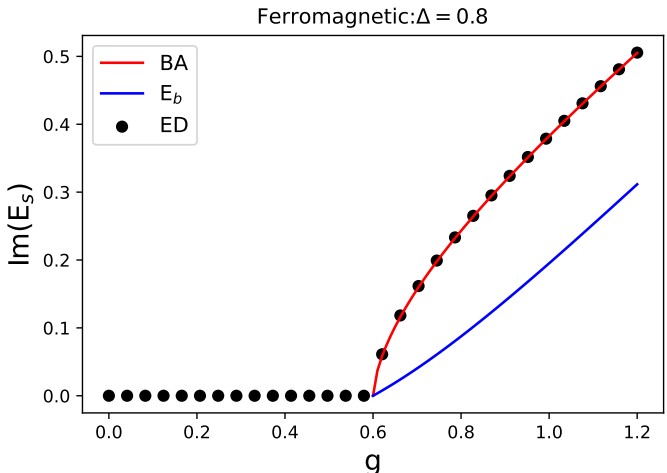

Figure 3: Imaginary part of the steady state energy $E_s$ as a function of the non-Hermitian parameter $g$. We take $\Delta = 0.8$, so that $g_c = 0.6$. Red curve is from the Bethe ansatz (BA). As a comparsion, blue line is imaginary energy of a boundary mode ($E_b$). The black dots are computed from exact diagonalization (ED) in the zero-magnetization subspace, with system size $N = 16$.

The summation of energy of all those scale-free modes becomes an integral over $\lambda$ in the thermodynamic limit. The imaginary part of energy formula of a single magnon is

$$\text{Im}(E(x)) = \frac{1}{N}E'(\lambda)\sigma(\lambda) = -\frac{\sin(\gamma)}{N\gamma}K_1'(\lambda)\sigma(\lambda). \tag{31}$$

Here, Eq. (25) has been used. While each one contributes $\mathcal{O}(1/N)$ to the total imaginary part, the sum of contributions from all scale-free magnons is comparable to the boundary mode contribution $E_b$:

$$
\begin{aligned}
\sum_{j\neq 0}\text{Im}(E(x_j)) &= -\frac{\sin(\gamma)}{\gamma}\int_0^{+\infty} d\lambda\rho(\lambda)K_1'(\lambda)\sigma(\lambda) \\
&= -\frac{\sin(\gamma)}{2\gamma}\int_{-\infty}^{+\infty} d\omega\, 2\pi i\omega\tilde{K}_1(\omega)\tilde{\sigma}_\rho(\omega) \\
&= -\frac{\sin(\gamma)}{2\gamma}\int_{-\infty}^{+\infty} d\omega\, i\omega\frac{\tilde{K}_1(\omega)}{1+\tilde{K}_2(\omega)}\Theta(\omega) \\
&= \frac{\sin(\gamma)}{\gamma}\int_0^{+\infty} d\omega \sin(\omega\lambda_0)\tanh(\omega)\frac{\sinh[(\frac{\pi}{\gamma}-2)\omega]}{\sinh(\frac{\pi}{\gamma}\omega)}.
\end{aligned}
\tag{32}
$$

The imaginary part of the steady-state eigen-energy is then given by adding $E_b$:

$$\text{Im}(E_s) = \sum_{j\neq 0}\text{Im}(E(x_j)) + E_b. \tag{33}$$

In Fig. 3, we compare our formula with the exact diagonalization (ED) results in the $M = N/2$ sector (zero magnetization sector), which agrees excellently. The boundary field $g$ controls the imaginary part of the energy totally via $\lambda_0 = \frac{1}{\gamma}\ln\frac{g-g_c}{g+g_c}$. As $g$ crosses $g_c$, the steady state energy becomes complex.

## 3.4 Boundary bound state at $\Delta > 1$ and phase transition

Anisotropic interaction $\Delta > 1$ prefers bounding all magnons together, and therefore the magnons in the steady state tend to localize to the boundary. Specifically, the first magnon is bound to the boundary, and the next one is bounded to the previous one recursively. In the context of integrable spin models, the bound state is named an "string"; we shall follow this terminology and call our bound state near the boundary a "boundary string". We note that similar states have been identified in the spin model subjected to a non-Hermitian magnetic field at only one end [26]. In the thermodynamic limit, Bethe roots $\{\beta_j\}$ satisfies a recursive relation:

$$\beta_{j+1} + \beta_j^{-1} = 2\Delta, \ \beta_1 = \Delta - ig. \tag{34}$$

The imaginary part of energy is given by

$$
\begin{aligned}
\mathrm{Im}(E_s) &= -\frac{1}{2}\sum_{j=1}^{M}\mathrm{Im}(\beta_j + \beta_j^{-1}) \\
&= -\frac{1}{2}\sum_{j=1}^{M-1}\mathrm{Im}(\beta_j^{-1} + \beta_{j+1}) - \frac{1}{2}\mathrm{Im}(\beta_1 + \beta_M^{-1}) \\
&= -\frac{1}{2}\mathrm{Im}(\beta_1 + \beta_M^{-1}).
\end{aligned}
$$

For large $N$, $\beta_{N/2}$ approaches the fixed point of recursive relations Eq. (34): $\beta_\infty = \Delta \pm \sqrt{\Delta^2 - 1}$, which is purely real. It follows that $\mathrm{Im}(E_s) = -\frac{1}{2}\mathrm{Im}(\beta_1) = \frac{g}{2}$.

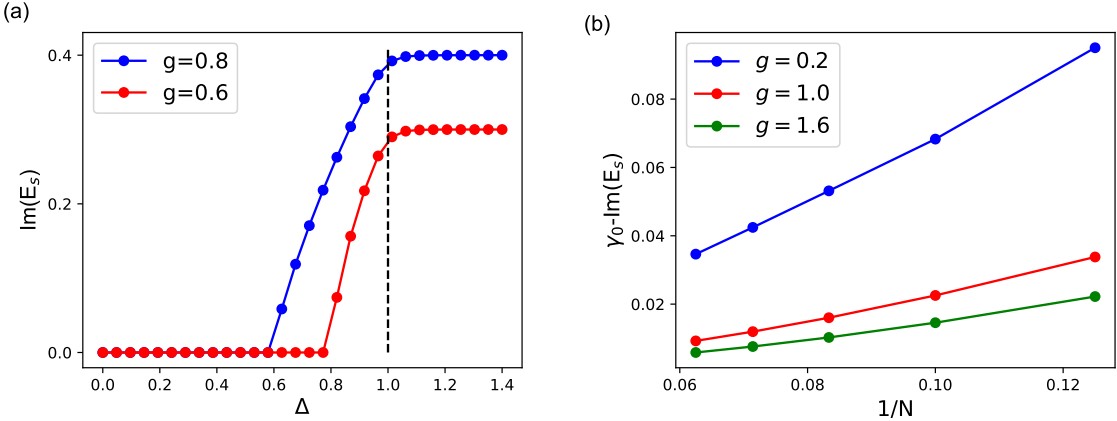

Figure 4: (a) The imaginary part of the steady state energy in the zero magnetization sector, which are obtained from exact diagonalization with system size $N = 16$. (b) Finite-size scaling of the steady-state energy at the isotropic point $\Delta = 1$. $\gamma_0 = g/2$ [see the paragraph above Eq. (36)].

It is clear that the structure of Bethe roots of the steady state is different for $0 < \Delta < 1$ and $\Delta > 1$. We may take the isotropic limit $\Delta = 1$ from the two sides to understand the phase transition point.

In the scale-free phase, we have to deal with the $\gamma \to \pi$ limit of Eq. (32) carefully. Note that the Fermi sea ranging from $-\pi + \gamma$ to $\pi - \gamma$ shrinks to a Fermi point when $\gamma \to \pi$. We take the limit by substituting $\omega$ by $\omega\gamma/\sin\gamma$ with $\sin\gamma \ll 1$, and the integral can be simplified to

$$\int_0^{+\infty} d\omega \sin(2\omega/g)\exp(-2\omega) = g/2(1+g^2), \tag{35}$$

so that $\text{Im}(E_s) = E_b + g/2(1 + g^2) = g/2$.

In the boundary-string phase, the imaginary part is a constant $\gamma_0 = g/2$. Bethe roots can be determined by the recursive equation Eq. (34) at $\Delta = 1$, which results in an explicit solution:

$$\beta_n = 1 + \frac{1}{n - 1 + \frac{i}{g}}. \tag{36}$$

For small $n$, the magnon localizes exponentially at the boundary. However, for $n$ sufficiently large ($n/N \sim \mathcal{O}(1)$), it behaves in a scale-free fashion. This result is also confirmed by comparing the imaginary part with $\gamma_0$ for different system sizes, and the differences scale linearly with $1/N$ (see Fig. 4). We emphasize that the solution Eq. (36), though qualitatively valid, is not exact because the scattering between the large $n$ scale-free modes have been neglected. This approximation has been implied in Eq. (34).

## 4   Ground state phase diagram

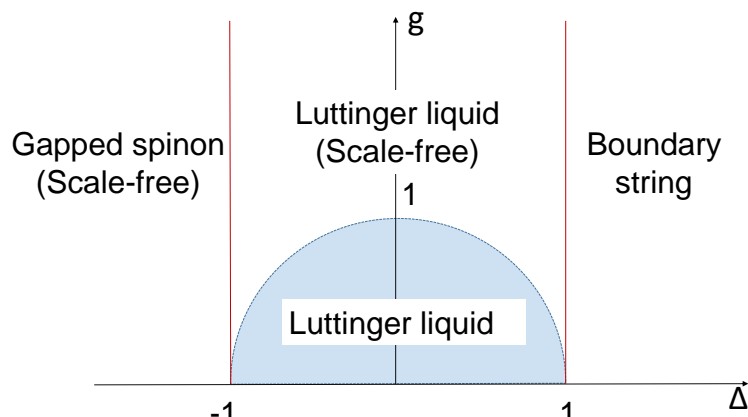

Figure 5: Ground state phase diagram in the zero magnetization sector. The right half of the diagram, $\Delta > 0$, coincides with the counterpart of steady state (see Fig. 1). In the $PT$ exact phase ($g < g_c \equiv \sqrt{1 - \Delta^2}$), the ground state is Luttinger liquid (LL); for $g > g_c$, gapless excitations become scale-free. When $\Delta < -1$, the gap between the ground state and excited states opens, yet our ansatz of scale-free skin modes remains valid.

For the ferromagnetic case $\Delta > 0$, the steady state and the ground state coincide, and the phase boundary $\Delta = 1$ separates the scale-free phase and the boundary string. However, the transformation $ZT$ relating the ferromagnetic and the anti-ferromagnetic steady state is not applicable to the ground state, because it changes to the highest-energy (real part) state when reversing the sign of $\Delta$. Therefore, one cannot borrow the ground-state phase diagram from that of the steady state. Moreover, the comparison between ground states at zero and finite non-Hermiticity provides another perspective on the effect of boundary dissipation. In Fig. 5, we summarize the results of the ground state. The ansatz of many-body scale-free state in the region $\Delta < 0$ is studied in the following subsections.

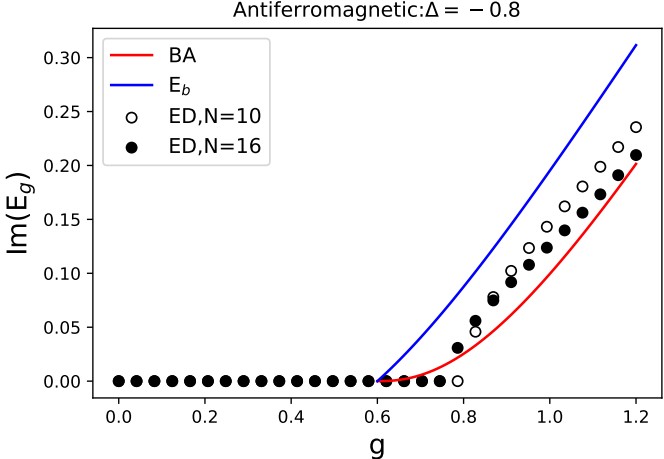

Figure 6: Imaginary part of the ground state energy $E_g$ as a function of the non-Hermitian parameter $g$. We take $\Delta = -0.8$, so that $g_c = 0.6$. Red curve is from the Bethe ansatz (BA). As a comparison, blue curve is imaginary part of the energy of a boundary mode ($E_b$). The hollow dots are obtained from exact diagonalization (ED) with $N = 10$, and solid dots with $N = 16$.

## 4.1 Scale-free solutions for $-1 < \Delta < 0$

For $-1 < \Delta < 0$, we find that Eq. (32) can still be applied to the ground state, though it does not coincide with the steady state anymore. The formula is compared with exact diagonalization results in Fig. 6. It seems that the data does not agree as well as in the ferromagnetic case (see Fig. 3). This is due to the larger finite-size error. In fact, we observe that as size $N$ increases, the ED results become closer to the analytical ones.

## 4.2 Imaginary Bethe equation at $\Delta < -1$

The imaginary Bethe equation also works for $\Delta < -1$, with some technical modifications. Retaining the parametrization $\Delta = -\cos(\gamma)$, $|\Delta| > 1$ now leads to a purely imaginary $\gamma$, and it is convenient to write $\gamma \to i\phi$:

$$\beta_j = -\frac{\sin[\phi(x_j - i)/2]}{\sin[\phi(x_j + i)/2]}, \tag{37}$$

where $\phi = \text{arccos h}(-\Delta)$. The boundary Bethe root is

$$x_0 = -\frac{2}{\phi}\arctan(\frac{\sinh(\phi)}{g}) - i = \lambda_0 - i. \tag{38}$$

On the ground state the whole Brillouin zone $k \in (-\pi, \pi)$ is filled, so that $\text{Re}(x) \in (-\pi/\phi, \pi/\phi)$. The single-magnon kinetic energy is

$$E(x_j) = \frac{1 - \cosh(\phi)\cos(\phi x_j)}{\cosh(\phi) - \cos(\phi x_j)}. \tag{39}$$

We also adopted here a new definition of the function $\theta_n$:

$$\theta_n(x) = 2\arctan[\coth(n\phi/2)\tan(\phi x/2)].$$

The imaginary Bethe equation is then given by

$$\rho(\lambda)\sigma(\lambda) + \int_{-\pi/\phi}^{+\pi/\phi} d\lambda' \frac{K_2(\lambda - \lambda')}{2\pi}\rho(\lambda')\sigma(\lambda') = \frac{1}{2\pi}\text{Im}[\theta_2(\lambda - x_0) + \theta_2(\lambda + x_0)]. \quad (40)$$

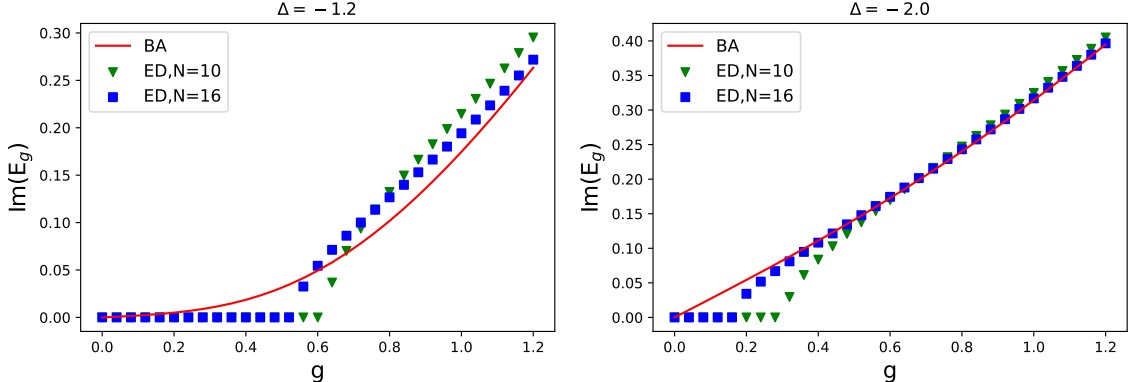

Figure 7: Imaginary part of the ground state energy as a function of the non-Hermitian parameter $g$. Left panel: $\Delta = -1.2$; Right panel: $\Delta = -2.0$. Red curve is obtained from the Bethe ansatz (BA). Green triangles and blue squares are numerical results from exact diagonalization of $N = 10$ and $N = 16$, respectively.

Since functions of $\lambda$ are periodic functions with periodicity $2\pi/\phi$, we can expand them as Fourier series to solve the integral equation:

$$\tilde{f}(m) = \int_{-\pi/\phi}^{+\pi/\phi} \frac{d\lambda}{2\pi} f(\lambda)e^{im\phi\lambda}, \; m \in \mathbb{Z} \quad (41)$$

Notably, $\tilde{K}_n(m) = \exp(-n|m|\phi)$, and the right hand side of the eq. (40) transforms as

$$\int_{-\pi/\phi}^{+\pi/\phi} \frac{d\lambda}{2\pi} e^{im\phi\lambda}\text{Im}[\theta_2(\lambda - x_0) + \theta_2(\lambda + x_0)]$$

$$= 2i\sin(m\phi\lambda_0)\int_{-\pi/\phi}^{+\pi/\phi} \frac{d\lambda}{2\pi}e^{im\phi\lambda}\text{Im}[\theta_2(\lambda + i)]$$

$$= \frac{2\sin(m\phi\lambda_0)}{m\phi}\int_{-\pi/\phi}^{+\pi/\phi} \frac{d\lambda}{2\pi}e^{im\phi\lambda}\frac{2\phi\cosh(\phi)\sinh^2(\phi)\sin(\phi\lambda)}{(\cos(\phi\lambda) - \cosh(\phi))(\cos(\phi\lambda) - \cosh(3\phi))}$$

$$= 2i\frac{\sin(m\phi\lambda_0)}{m\phi}\sinh(m\phi)e^{-2m\phi} = \Theta(m). \quad (42)$$

The imaginary part of energy is:

$$\sum_{j\neq 0}\text{Im}(E(x_j)) = -\frac{\sinh(\phi)}{\phi}\int_0^{+\pi/\phi} d\lambda\rho(\lambda)K_1'(\lambda)\sigma(\lambda)$$

$$= -\frac{\sinh(\phi)}{2}\sum_{m\in\mathbb{Z}} 2\pi i m\phi\tilde{K}_1(m)\tilde{\sigma}_\rho(m)$$

$$= -\frac{\sinh(\phi)}{2}\sum_{m\in\mathbb{Z}} im\phi\frac{\tilde{K}_1(m)}{1 + \tilde{K}_2(m)}\Theta(m)$$

$$= \sinh(\phi)\sum_{m\in\mathbb{Z}_+}\sin(m\phi\lambda_0)\tanh(m\phi)e^{-2m\phi}. \quad (43)$$

As illustrated in Fig. 7, there are finite-size errors between the numerical results and our Bethe ansatz formula, which is similar to the gapless antiferromagnetic case.

## 5 Experimental Scheme

The onsite non-Hermiticity can be realized in cold atom systems by coupling the spin down (up) degrees of freedom of the first (last) site to an auxiliary state by optical pumping [59,60]. Each atom in the bulk has two effective energy levels to mimic a 1/2 spin. XXZ interaction can be induced by the mixing of even and odd parity states, e.g. in Rydberg atoms [61]. We may introduce the third energy levels on the two ends so that the effective spin down (up) state on the left (right) end can decay to it spontaneously. The effective loss is described by a non-Hermitian term

$$H_{\text{loss}} = -\frac{ig}{2}|\uparrow\rangle\langle\uparrow|_1 - \frac{ig}{2}|\downarrow\rangle\langle\downarrow|_N = -\frac{ig}{2} + \frac{ig}{2}(S_N^z - S_1^z).$$

To evolve the open system under the non-Hermitian Hamiltonian without quantum jump, i.e., by post-selection, the population of auxiliary energy levels should be monitored by exciting the states with laser. The absence of fluorescence signals the absence of the quantum jump from the magnetization-conserved quantum trajectory. The evolution of the many-body state is then governed by the effective Hamiltonian $H_{\text{eff}} = H_{\text{XXZ}} + H_{\text{loss}}$, which differs from our initial Hamiltonian Eq. (1) only by an imaginary constant $\frac{ig}{2}$.

During the time interval $[t, t + \delta t]$, the state $|\psi\rangle$ evolves as

$$|\psi(t+\delta t)\rangle = \frac{\exp(-iH_{\text{eff}}\delta t)|\psi(t)\rangle}{|\exp(-iH_{\text{eff}}\delta t)|\psi(t)\rangle|},$$

Starting from an initial state in the zero magnetization sector, the system will relax to the steady state after sufficiently long time. The imaginary part of the corresponding eigenvalue can be obtained by measuring the expectation of boundary spin polarization:

$$\text{Im}(E_s) = \text{Im}\langle\psi_s|H_{\text{eff}}|\psi_s\rangle + \frac{g}{2} = \frac{g}{2}\langle\psi_s|(S_N^z - S_1^z)|\psi_s\rangle = g\langle\psi_s|S_N^z|\psi_s\rangle, \tag{44}$$

where $|\psi_s\rangle$ is the steady state. The measured boundary spin polarization can be compared to our analytical result of $\text{Im}(E_s)$.

Numerical simulations for post-selection evolution under $H_{\text{eff}}$ are conducted on $N = 14$ chain to back up the above proposal. We consider two kinds of initial states. The first one is a "local quench", in which the spin chain is prepared in the ground state of $H_{\text{XXZ}}$, and boundary coupling to the auxiliary energy levels is turned on at certain moment. The other initial state is a domain-wall configuration, in which the spins of the left half chain point down while those of the right half point up. We discretize the continuous time evolution by fourth order Runge-Kutta method, and obtain the spin polarizations in Fig. 8(a-d). It is clear that for both local quench and domain wall initial states, the edge spin polarization converges to the predictions of Bethe ansatz solution. For certain parameter choices, e.g., Fig. 8(b), the relaxation time towards the steady state is comparable to $1/g$, which is most convenient from experimental perspective. The results from Bethe ansatz and numerical simulation are also confirmed by exact diagonalization Fig. 8(e)(f). Fig. 8(e)(f) also shows clearly that the steady-state expectation of $S_N^z$ is well below 1/2 for $|\Delta| < 1$ , while it saturates to 1/2 for $|\Delta| > 1$.

## 6 Conclusion

In this work, we applied coordinate Bethe ansatz to solve the steady state and the ground state of a $PT$ symmetric one-dimensional boundary-dissipated spin chain, focusing on the $PT$

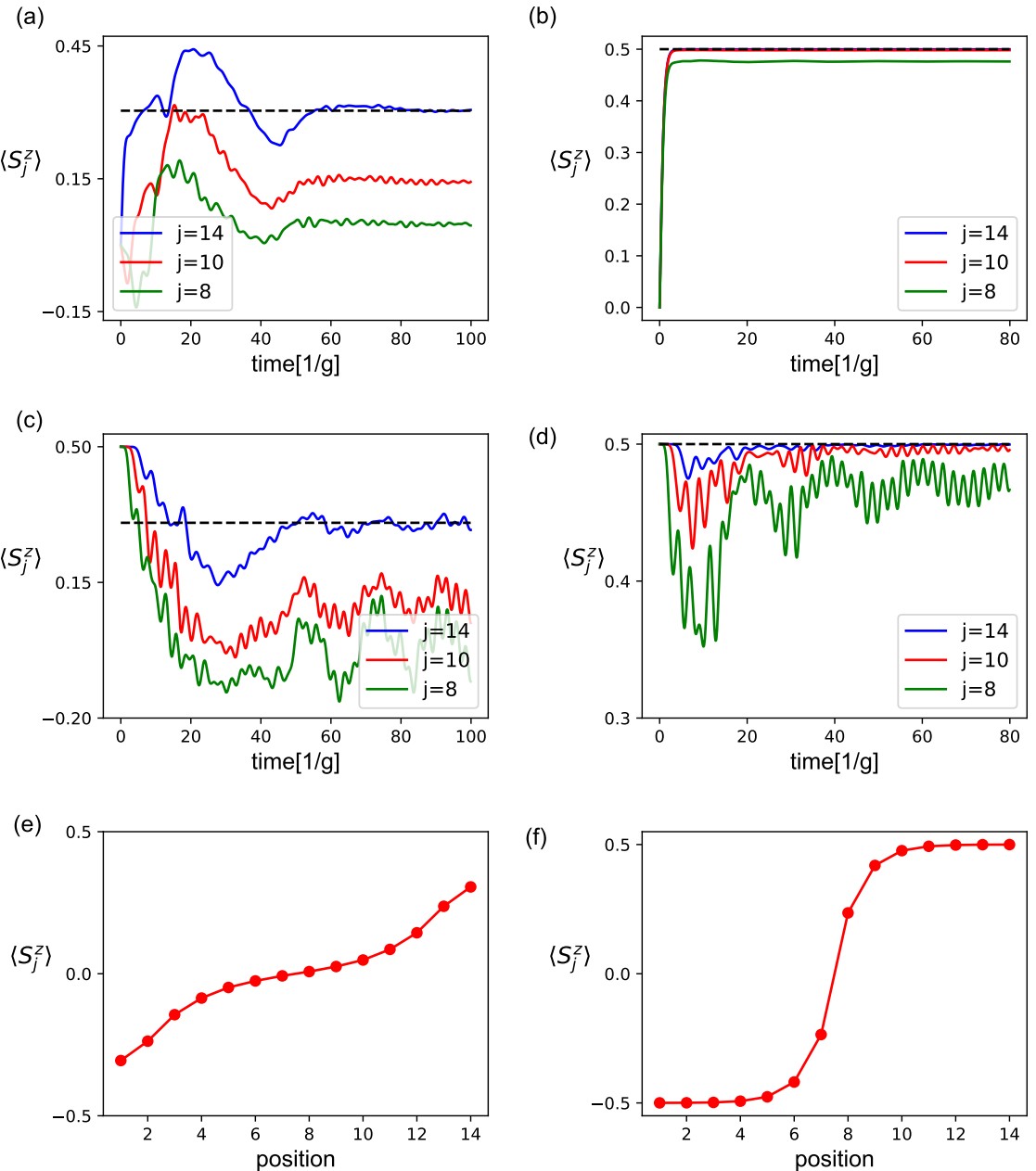

Figure 8: (a)-(d) Time evolution of spin polarization under post-selection dynamics. The dashed lines indicate the values of $\frac{\text{Im}(E_s)}{g}$ obtained from the Bethe ansatz. For (a)(b), the time evolution starts from a local quench; in (c)(d), it starts from the "domain-wall" initial state (see text). The time is measured in unit of $1/g$. (e)(f) Steady-state spin polarization profile obtained from exact diagonalization. Parameter values: $N = 14$ and $g = 0.8$ are fixed. For (a)(c)(e), $\Delta = 0.8$; for (b)(d)(f), $\Delta = 1.2$.

broken phase. We found the many-body scale-free state, which is composed of one boundary mode and a continuum of scale-free modes in our particular model. We then derived the Bethe equations of the scale-free Bethe roots, and obtained a compact formula for the eigen-energy in the thermodynamic limit. We then proposed an experimental scheme to measure the dissipative part of the energy, and discussed how to compare it with our analytical results.

Our findings shed a light on exceptional points and $PT$ transition in many-body physics. Particularly, our solution is a generalization of the concept of scale-free NHSE from free-particle to many-body systems. Although we focused on the scale-free behaviour in the XXZ spin chain, it is expected that this feature is universal in a family of non-Hermitian models with interactions in the bulk and dissipation-induced defect mode at the boundary. For example, non-integrable models also exhibit scale-free properties, as demonstrated in Fig. 2. Moreover, in integrable models solvable by nested Bethe ansatz (Fermi-Hubbard model, higher spin XXX chain, etc.), boundary-operator induced $PT$ symmetry transition and the corresponding steady states may have richer structures to uncover.

We have demonstrated the scale-free skin effect by the difference between the imaginary part of the steady state energy and that of a single boundary mode. Intuitively, the scale-free skin effect may also manifest itself in the excitation spectra near the steady state. A thorough analysis of those eigenstates requires preserving the $\mathcal{O}(1/N^2)$-order terms in the imaginary Bethe equations, which is left for future studies. Algebraic Bethe ansatz is another possible approach to the solutions of excitations, though it remains a question how scale-free Bethe roots emerge in the monodromy and transfer matrices.

# Acknowledgements

This work is supported by NSFC under Grant No. 12125405.

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
