# Peer review of "Scale-free non-Hermitian skin effect in a boundary-dissipated spin chain"

_SciPost Physics_

## Round 1 · Referee Report · Anonymous (Referee 1) · 2023-7-2

Strengths

1- Study of the non Hermitian skin effect in an interacting integrable XXZ spin chain

Weaknesses

1- The existing literature is insufficiently documented

Report

In this paper, the XXZ model with imaginary diagonal boundary fields is studied. The focus is on large enough boundary fields, for which the appearance of a scale-free skin effect is demonstrated. The authors also argue that the phenomenon occurs even for non integrable deformations.

Overall the results presented are interesting, even though the integrability techniques used here are well known. Since the study of non hermitian spin chains and skin effects has become quite topical of late, the authors findings are also timely.

I believe this deserves publication in some form, especially since most works on skin effect do not consider interactions. There are nevertheless some issues which are listed below. Provided those are addressed, the paper might be publishable in Scipost Physics.

1) The existing literature is insufficiently documented. For example, it is claimed that 'the unit circle $\Delta^2+g^2=1$ has been the focus in previous literatures', which is not quite true. For instance reference 1 does write the Bethe equations in the more general case, which the authors use them later on. There are also many works which study those open chains with even more general complex and not necessarily diagonal boundary terms. An effort should be made to read through this literature, and better explain how this works differs from previous ones. Other authors (some in the same Institute) have also recently looked at the skin effect in the integrable Lieb liniger model, see 2207.12637.

2) Page 5, 'Although it remains unclear whether all eigenstates of the non-Hermitian XXZ model have the form of Bethe ansatz'. Related to the previous point, there are works which address such issues, see for example 1401.4901.

3) The name imaginary Bethe equation is a little bit strange. This looks more like an imaginary Lieb equation, since Bethe equations usually refers to the discrete set of equations the authors start from.

4) Related point, in figures 3, 6, 7. I assume the authors refer to their integral equation rather than Bethe Ansatz, and compare to exact diagonalization. They could also compare with a numerical solution to the discrete Bethe equations. This would allow to reach greater system sizes, and show more convincingly the agreement with the continuum equation.

I also noticed the following minor issues:

5) Page 2: 'coordinated Bethe ansatz' $\to$ 'coordinate Bethe ansatz' and 'proportional inversely' $\to$ 'inversely proportional'

6) Figure 3. 'As a comparsion' $\to$ 'As a comparison'

7) The page break on page 16 could be avoided.

Requested changes

See above.

  • validity: high
  • significance: high
  • originality: high
  • clarity: high
  • formatting: excellent
  • grammar: good

Author:  Heran Wang  on 2023-08-24  [id 3922]

(in reply to Report 1 on 2023-07-02)
Category:
answer to question

We sincerely thank the referee for his/her time on reviewing the manuscript, and his/her invaluable and helpful suggestions and comments. We take the referee's questions very seriously and have conducted additional numerical calculations. We directly solve the Bethe equations with a larger length to compare with the analytical solutions. The detailed response is provided below.

1)"The existing literature is insufficiently documented. For example, it is claimed that 'the unit circle Δ^2+g^2=1 has been the focus in previous literatures', which is not quite true. For instance reference 1 does write the Bethe equations in the more general case, which the authors use them later on. There are also many works which study those open chains with even more general complex and not necessarily diagonal boundary terms. An effort should be made to read through this literature, and better explain how this works differs from previous ones."

Response: In the second paragraph of Introduction, we have added some sentences to discuss the previous literature on exact solutions of open spin chains with arbitrary boundary fields, where related references [26-28] are added. We point out that before our work, a general method to identify the steady state and the ground state for such models was lacking.

"Other authors (some in the same Institute) have also recently looked at the skin effect in the integrable Lieb liniger model, see 2207.12637."

Response: We added Ref. [49] about the non-Hermitian Lieb-Linger model the referee mentioned, and Ref [50] about the unidirectional Bose Hubbard model. We have clarified the differences between our boundary-dissipated model and the two models with non-Hermiticity adding in the bulk.

2)"Page 5, 'Although it remains unclear whether all eigenstates of the non-Hermitian XXZ model have the form of Bethe ansatz'. Related to the previous point, there are works which address such issues, see for example 1401.4901."

Response: We have included this paper in Ref. [58], and modified our expression, that "we specifically obtain the closed-form expression for the complex momentum distribution on the steady state and the ground state in the thermodynamic limit."

3)"The name imaginary Bethe equation is a little bit strange. This looks more like an imaginary Lieb equation, since Bethe equations usually refers to the discrete set of equations the authors start from."

Response: We agree that the name "imaginary Bethe equation" is not appropriate. However, we find that the "Lieb equation" always specifically refers to the integral equations for the density of Bethe roots in the Lieb-Linger model. Therefore, in the revised version we have adopted a more general name "imaginary Fredholm equation" for such inhomogeneous linear integral equations. We have also added the note Ref. [51] for the demonstration.

4) "Related point, in figures 3, 6, 7. I assume the authors refer to their integral equation rather than Bethe Ansatz, and compare to exact diagonalization. They could also compare with a numerical solution to the discrete Bethe equations. This would allow to reach greater system sizes, and show more convincingly the agreement with the continuum equation."

Response: We agree that we compared the solutions of the integral equation, instead of the discrete Bethe equations, with results from exact diagonalization. In response to your insightful suggestions, we have directly solved the discrete Bethe equations. The Bethe equations with open boundary conditions invlove high order [2(M+N)] algebraic equations with M variables. The crucial point to correctly solving the ground state Bethe roots is to identify an appropriate starting point for the solver to find roots. This step is particularly critical when dealing with systems of large length. Due to the small spacing between Bethe roots in such cases, an unsuitable starting point can potentially result in producing the Bethe roots far away from the ground state.

Here, we develop the adiabatic path method to find the ground state Bethe roots. Given the parameters (Δ_final, g_final), We initiate the process of solving from the non-interacting limit at (0, g_initial) with g_initial>1, with the presence of the boundary mode. At this point the Bethe equations are reduced to one 2N+2-order equation, allowing us to solve all the Bethe roots with high precision and manually select N/2 roots to occupy the ground state. Next, we incrementally increase the anisotropy Δ in small steps (e.g. 0.01), and for each step, we search the roots around the solutions for one step before. This iterative procedure continues until Δ reaches the desired value Δ_final. After that, we adopt a similar method to smoothly approach g_final by increasing (or decreasing) the field g. We follow an adiabatic path connecting the initial point (0, g_initial) to the target point (Δ_final, g_final), and maintain proximity to the ground-state solutions throughout the process.

In the gapless phase (|Δ|<1), we have applied the method and solved the discrete Bethe equations at Δ=-0.8 for length N=40, and added the data in Fig. 6 (attached below). The data matches the results from integral equation (the red line) better, in comparison to the small-size exact diagonalization. Promisingly, the numerical points for g>1 effectively converge towards the analytical line, which is not reached by the exact diagonalization. We believe that the inclusion of this new dataset provides more convincing evidence for supporting the validity of the imaginary Fredholm equation in our work.

The adiabatic path method will be invalid when the path crosses the phase boundary, i.e. Δ^2+g^2=1 or Δ=± 1, since then the significant fluctuations around the phase boundary can make the solver miss the goal. Consequently, starting from the non-interacting limit, we can never approach the gapped phase with |Δ|>1. Therefore, we have not applied the method to the parameters of Fig. 7.

The above procedure of calculations has been also included in the appendix of our manuscript.

5) 6)7) "Page 2: 'coordinated Bethe ansatz' →'coordinate Bethe ansatz' and 'proportional inversely'→'inversely proportional' Figure 3. 'As a comparsion' →'As a comparison' The page break on page 16 could be avoided."

Response: Thanks to the referee for reminding us of the typos, and we have fixed all of them.

Attachment:

BAED_anti_reply.pdf

---

## Round 1 · Referee Report · Anonymous (Referee 2) · 2023-7-12

Strengths

  1. Provides a novel link between two different research areas: skin effects in non-hermitian quantum systems and Bethe Ansatz

  2. Interesting phase diagrams (figs. 1 and 5) supported by exact results.

Weaknesses

  1. The bibliography could be more complete. In particular, the Bethe Ansatz solution is not really new, compare for instance the Bethe equations (Eq. (16)) to Eq. (3.27) in [N Kitanine et al J. Stat. Mech. (2007) P10009].

  2. The relevance of some results is unclear, probably because they are insufficiently discussed. For instance, the authors should elaborate on the effects of the integrability-breaking perturbation at the end of Sec. 3.2, and on the numerical results displayed in Fig. 8 about post-selected quench dynamics.

Report

The authors present a thorough study of the XXZ spin chain with imaginary (non-hermitian) diagonal boundary terms, motivated by recent works on skin effects in non-hermitian systems. While most of these recent works have focused on non-interacting fermion systems, the present paper studies the interacting XXZ chain, and brings the tools of coordinate Bethe Ansatz into the game. I believe this paper has the potential to provide a link between these two subfields of physics. It also presents interesting findings, such as the phase diagrams in figs. 1 and 5. Therefore I think the paper deserves to appear in Scipost Physics.

Requested changes

Some parts of the paper could be improved. Some of the figures are shown almost without comments. For instance, why are the results about integrability breaking important? They are shown in Fig. 2, but almost nothing is said about them in the text, apart from two sentences around Eq. (10). Similarly, what is the physical meaning of the results shown in Fig. 8? All the figures should be discussed properly.

Also, the bibliography could be more complete, and the connection between this paper and previous Bethe Ansatz literature should be made more explicit. Relevant references such as [N Kitanine et al J. Stat. Mech. (2007) P10009] seem to be missing.

Finally, the comment about "recently introduced non-unitary CFTs [31,32]" in the introduction sounds really wrong: the existence of non-unitary CFTs has been known since the early days of 2d CFT, see e.g. Ginsparg's lecture notes (arXiv:hep-th/9108028), and many non-unitary CFTs have been studied in statistical mechanics, for instance as the continuum limit of loop models (see e.g. [Foda and Nienhuis, Nuclear Physics B, 324(3), 643-683, 1989]).

  • validity: high
  • significance: high
  • originality: high
  • clarity: high
  • formatting: good
  • grammar: excellent

Author:  Heran Wang  on 2023-08-24  [id 3923]

(in reply to Report 2 on 2023-07-12)
Category:
answer to question
correction

We sincerely thank the referee for his/her time on reviewing the manuscript, and his/her thoughtful comments and efforts towards improving our manuscript. The detailed response is provided below.

1)"Some of the figures are shown almost without comments. For instance, why are the results about integrability breaking important? They are shown in Fig. 2, but almost nothing is said about them in the text, apart from two sentences around Eq. (10). "

Response: In Sec. 3.2, below Eq. (11), we have added a paragraph to discuss the integrability breaking terms, and pointed out that the qualitative behaviour of the scale-free modes is still captured by Eq. (9) and (10).

"Similarly, what is the physical meaning of the results shown in Fig. 8? All the figures should be discussed properly."

Response: We have added a paragraph in Sec. 5, where we provided a thorough analysis of Fig. 8, and discussed relations between the simulation data and our analytical solutions in the above sections.

2) "Also, the bibliography could be more complete, and the connection between this paper and previous Bethe Ansatz literature should be made more explicit. Relevant references such as [N Kitanine et al J. Stat. Mech. (2007) P10009] seem to be missing."

Response: In the second paragraph of Introduction, we have added some sentences to discuss the previous literature on exact solutions of open spin chains with arbitrary boundary fields, where related references [26-28] are added. The recommended paper (Ref. [7]) is included in the first paragraph as an example of spin chains with diagonal boundary field.

3)"Finally, the comment about "recently introduced non-unitary CFTs [31,32]" in the introduction sounds really wrong: the existence of non-unitary CFTs has been known since the early days of 2d CFT, see e.g. Ginsparg's lecture notes (arXiv:hep-th/9108028), and many non-unitary CFTs have been studied in statistical mechanics, for instance as the continuum limit of loop models (see e.g. [Foda and Nienhuis, Nuclear Physics B, 324(3), 643-683, 1989])."

Response: In the second paragraph of Introduction, we have modified the sentence about the non-unitary CFT. Concretely, we have added Ref. [33,34] for introducing the non-unitary CFT, and referred the papers Ref. [34,35] as the entanglement criterion of the non-Hermitian XXZ chain to identify the model as the non-unitary CFT.

---

## Editorial Decision

resubmitted